# Kombucha: Production and Microbiological Research [note 1]

**DOI:** 10.3390/foods11213456

**Published:** 2022-10-31

**Authors:** Boying Wang, Kay Rutherfurd-Markwick, Xue-Xian Zhang, Anthony N. Mutukumira

**Affiliations:** 1School of Food and Advanced Technology, Massey University, Auckland 0745, New Zealand; 2School of Health Science, College of Science, Massey University, Auckland 0745, New Zealand; 3School of Natural Sciences, Massey University, Auckland 0745, New Zealand

**Keywords:** Kombucha, fermentation, acetic acid bacteria, yeast, microbial identification

## Abstract

Kombucha is a sparkling sugared tea commonly prepared using a sugared tea infusion and fermented at ambient temperature for several days using a cellulose pellicle also called tea fungus that is comprised of acetic acid bacteria and yeast. Consumption of Kombucha has been reported as early as 220 B.C. with various reported potential health benefits and appealing sensory properties. During Kombucha fermentation, sucrose is hydrolysed by yeast cells into fructose and glucose, which are then metabolised to ethanol. The ethanol is then oxidised by acetic acid bacteria (AAB) to produce acetic acid which is responsible for the reduction of the pH and also contributes to the sour taste of Kombucha. Characterisation of the AAB and yeast in the Kombucha starter culture can provide a better understanding of the fermentation process. This knowledge can potentially aid in the production of higher quality products as these microorganisms affect the production of metabolites such as organic acids which are associated with potential health benefits, as well as sensory properties. This review presents recent advances in the isolation, enumeration, biochemical characteristics, conventional phenotypic identification system, and modern genetic identification techniques of AAB and yeast present in Kombucha to gain a better understanding of the microbial diversity of the beverage.

## 1. Introduction

Kombucha is a traditional fermented sparkling tea beverage with a slightly sweet and acidic flavour that has been consumed in China since around 220 B.C. The beverage is consumed for its refreshing sensory characteristics and the perceived inherent health-promoting properties [1]. It is reported that Dr Kombu introduced fermented tea to Japan around 414 A.D., where it was apparently used to alleviate digestive ailments [1,2]. The name, ‘Kombucha’ is documented to originate from “Dr Kombu” and “cha” refers to tea in Japanese. Kombucha was introduced to Russia as “Tea Kvass” and then it spread to Eastern Europe during the 20th century. The popularity of Kombucha in Russia was attributed to its purported beneficial effects on healing metabolic diseases, haemorrhoids and rheumatism [3,4]. During World War II, Kombucha was introduced to Western Europe and North Africa. In recent decades, the production of Kombucha has become global, and it is now sold as a commercial beverage with different flavours. In addition, due to the high popularity of Kombucha products, small-scale home-brewed products are often made for personal use and can frequently be found for sale at farmer’s markets and in communities.

The global market size for Kombucha has increased significantly in recent years. In 2018, Kombucha was worth USD 1.5 billion, and it is estimated to grow to around 5 billion by 2025 with an estimated compound annual growth rate (CAGR) of 23% [5]. Most commercial Kombucha products sold in New Zealand are flavoured using fruits and/or herbs in single or mixed flavours. Information on commercial Kombucha products from different brands sold in New Zealand is shown in Table 1.

Kombucha is usually fermented from sweetened tea using a symbiotic culture of bacteria and yeast which is commonly abbreviated to SCOBY. The SCOBY is also known as tea fungus, cellulosic pellicle or consortium. The microbial community in Kombucha is diverse and varies between fermentations, but it is mainly composed of AAB and yeast, although, the presence of small amounts of lactic acid bacteria (LAB) in Kombucha has also been reported [6]. These microorganisms (LAB, AAB and yeast) have been suggested to be potential probiotics and therefore contribute to the health benefits of Kombucha [7]. However, the probiotic performance of the live cultures identified in Kombucha has rarely been studied [8]. Therefore, the probiotic potential of the microorganisms found in Kombucha should be determined using both in vitro and in vivo studies.

Many different substrates can be used for Kombucha, including green tea, oolong tea and black tea, as well as medicinal herbs including lemon balm, peppermint, thyme and sage or combinations thereof [9]. Black tea is the most common tea substrate for Kombucha fermentation [10], and sucrose is the most popular source of carbon during fermentation [2]. For Kombucha fermentation, approximately 50 to 150 g/L (5% to 15% *w*/*v*) of sucrose is dissolved in boiled water, then tea leaves or tea bags are added to the sugared hot water and allowed to infuse for around 5–10 min, followed by filtration to remove tea leaves (or removal of tea bags). Sugared tea is allowed to cool to room temperature [1]. To prevent contamination from pathogenic and spoilage microorganisms, the production process is generally carried out under highly sanitary conditions with a small portion of previously fermented Kombucha broth added to reduce the pH [2,3]. The cooled sugared tea is then transferred to a sterile wide-mouth container, and the “mushroom” (tea fungus) is placed onto the surface of the solution. The vessel is covered with a sterile cloth or paper towel to prevent insects and other undesirable cross-contamination from affecting the fermentation. The fermentation generally takes 7 to 10 days at room temperature (20 °C to 30 °C), with fermentation temperatures ranging from 18 °C to 26 °C reported as optimal [2]. During fermentation, a newly formed jelly-like daughter tea fungus membrane is formed which floats on the surface of the broth. The tea fungus is removed and retained, together with a small amount of tea broth for the next fermentation. The daughter tea fungus can grow up to 2 cm thick and covers the surface of the mother tea fungus to gain better access to oxygen [2,11,12]. After fermentation, the liquid broth is filtered through a clean cloth and stored in a sealed container at 4 °C for further processing such as packaging. With fermentation, the taste of Kombucha changes from pleasantly fruity at the beginning, to a vinegar-like flavour after longer fermentation [2].

The consumption of Kombucha has been suggested to confer numerous health benefits for humans. However, the evidence for most of these benefits is based on in vitro studies. Therefore, in vivo clinical trials are required to demonstrate any biological functions of Kombucha and to correlate them with active compounds [13]. The fermented beverage is perceived to lower blood pressure (antihypertensive) by inhibiting the angiotensin-converting enzyme (ACE) and mediating blood sugar levels (antidiabetic) and cholesterol levels [14]. Furthermore, the consumption of Kombucha may be effective in weight management by controlling appetite due to its hypolipidemic effects associated with lipase inhibition [14,15]. Kombucha has been reported to exhibit anticarcinogenic activity due to the presence of tea polyphenols and metabolites [2]. The antimicrobial activity of Kombucha against pathogenic microorganisms is mainly attributed to the action of acetic acid produced during fermentation [14,16]. The hepatoprotective activity of Kombucha is mainly due to the presence of the potential detoxifying agent, D-saccharic acid 1.4-lactone (DSL). However, there is scanty information on the potential toxicity of Kombucha associated with metabolic acidosis and hyponatremia [2,4]. Thus, the recommended daily consumption of Kombucha should be no more than 118 mL (4 fluid ounces) for healthy people [13]. 

The objective of this review is to document the recent advances in the microbiological characteristics, composition, and phenotypical and molecular identification techniques of Kombucha. Systematic research is recommended as the identification of the microbial characteristics of Kombucha may allow producers to effectively control the fermentation process for the production of safer, consistently high-quality products.

## 2. Microbiological Characteristics of Kombucha

The Kombucha microbial community can be classified into two parts, that found in the floating cellulosic biofilm and that in the liquid broth [17]. An overview of the metabolic activities of the Kombucha fermentation process is summarised in Figure 1 [18]. Sucrose is hydrolysed by yeast cells into fructose and glucose, which are metabolised by yeast to produce ethanol and carbon dioxide. Glycerol can also be produced by yeast due to the high osmotic pressure and further oxidised by AAB to dihydroxyacetone (DHA) [15]. Some esters are also produced during this process, which contribute to the aroma development of Kombucha. Fructose is preferably utilised as the substrate rather than glucose [19], with the resulting ethanol produced being further metabolised by AAB to produce acetic acid, thereby reducing the ethanol content in the Kombucha. The low ethanol concentration facilitates the formation of the cellulosic pellicle [15]. Glucose is metabolised by AAB into gluconic and glucuronic acids. Autolysis of yeast provides vitamins and other nutrients to support the growth of the AAB [16]. Yeast autolysis is commonly detected during the maturation of alcoholic beverages, which may impact the aroma and flavour of Kombucha products.

### 2.1. Presence of Acetic Acid Bacteria in Fermented Foods

#### 2.1.1. AAB Used in the Production of Fermented Products

Acetic acid bacteria play an important role in the fermented food and beverage industry. These bacteria are widespread in the environment and can be found in sugary or acidic substances such as fruits and flowers [20]. The genus *Gluconobacter* is easily isolated from environments abundant in sugar, while the genera *Acetobacter* and *Gluconacetobacter* are found in alcoholic environments [15]. AAB are widely used in the production of fermented food and beverage products such as vinegar, lambic beers, red wine, cocoa, and nata de coco (an edible bacterial cellulose formed from AAB in coconut water and kefir). The genera *Acetobacter* and *Gluconacetobacter* prefer to oxidise ethanol rather than glucose, while the genus *Gluconobacter* oxidises glycerol and glucose [15].

#### 2.1.2. Dominant Acetic Acid Bacteria Found in Kombucha

Acetic acid bacteria and gluconic acid-producing bacteria are the dominant prokaryotes found in Kombucha starter cultures. AAB belong to the family *Acetobacteraceae* and are classified into acetous or acidophilic groups. Currently, AAB are divided into 17 genera, of these, *Acetobacter*, *Gluconobacter*, *Gluconacetobacter* and *Komagataeibacter* are mainly used in the food industry [20,21]. The microbial community in Kombucha varies between products, which impacts the biochemical properties of the fermented beverage [22]. The production of the cellulosic biofilm (tea fungus) floating on the surface of the tea broth in Kombucha is associated with the presence of the AAB species *Komagataeibacter (K.) xylinus* [1,2,16,23]. Synthesis of the cellulosic pellicle involves the production of the cellulose precursor-uridine diphospho-glucose (UDPGc), which can be synthesised from several carbon sources including ethanol, sucrose, fructose, and glycerol [16]. The presence of caffeine, theophylline and theobromine facilitates the production of the cellulose network by *K. xylinus* [24,25], which is also affected by sugar types, sugar concentration, and pH [15].

Other AAB isolated from Kombucha include *Bacterium (B.) gluconicum* [10,26], *Acetobacter (A.) aceti*, *Acetobacter (A.) pasteurianus*, *Glucobacter (G.) oxygendans* [27], *Acetobacter (A.) musti* and *Gluconobacter (G.) potus* [28]. Recently, species of *Komagataeibacter* and *Gluconoacetobacter* have been reported in Kombucha including *Komagaeibacter (K.) kombuchae* [29], *Komagataeibacter (K.) saccharivorans* [30], *Komagataeibacter (K.) rhaeticus* [31], *Gluconacetobacter (G.) sacchari* [32] and *Gluconacetobacter* sp. A4 (G. sp. A4) are the main functional AAB isolated from Kombucha which synthesise D-saccharide acid-1,4 lactone (DSL), a promising detoxifying and antioxidant agent [33]. Nitrogen-fixing *Acetobacter (A.) nitrogenifigens* sp. nov. and *Gluconactobacter (G.) kombucahe* sp.nov have also been isolated from Kombucha [34].

### 2.2. Lactic Acid Bacteria Isolated from Kombucha

LAB have been utilised in different food products to achieve a unique flavour and deliver health benefits [35]. The genera *Lactobacillus* and *Bifidobacterium* are commonly isolated from fermented foods [36]. LAB may be present in Kombucha; however, current information is not considered essential for its production. In certain Kombucha products, LAB have been reported to make up as much as 30% of the bacterial community. The addition of some LAB strains—*Lacticaseibacillus (L.) casei* and *Lactiplantibacillus (L.) plantarum* have been shown to enhance the antioxidant and antimicrobial activities of Kombucha [35]. Moreover, the presence of LAB during Kombucha fermentation has been reported to enhance the production of D-saccharic acid, 1,4 lactone (DSL) and glucuronic acid [33]. DSL has detoxifying properties and may be an important factor contributing to the purported hepatoprotective activity of Kombucha [37]. The LAB isolated from Kombucha are shown in Table 2.

### 2.3. Yeast Isolated from Kombucha

#### 2.3.1. General Characteristics of Yeast

Yeast is a group of eukaryotic unicellular fungi which reproduce by budding or fission, with some capable of both [41]. Yeast has been used for the fermentation of foods and beverages for thousands of years due to its ability to hydrolyse different substrates to produce valuable fermented final products such as beer, wine and bread [42]. The first yeast used commercially was identified as the *Saccharomyces* (*S*.) *sensu stricto* complex [42]. Yeast is currently classified into Ascomycetous and Basidiomycetous yeast, according to their molecular phylogenies [41]. Yeasts are facultative anaerobes indicating that they can grow without oxygen. In the presence of oxygen, yeasts convert sugars to carbon dioxide and energy, whereas under anaerobic conditions, the sugars are converted to ethanol, glycerol and carbon dioxide [43].

*S. cerevisiae*, also known as baker’s yeast, is the most common food-grade yeast, widely used in bakery products, beer-brewing, and winemaking. Food-grade yeasts are also utilised as food additives, flavouring agents and as substrates for microbiological agar [43]. Some non-*Saccharomyces* yeast such as the genera *Candida*, *Debaryomuces*, *Kluyveromyces*, *Yarrowia* and *Zygosaccharomyces* are gaining attention in industry for their possible commercial applications as starter cultures for food and non-food products due to their perceived probiotic properties and impact on the development of flavour and colour in some meat products [42,44].

#### 2.3.2. Dominant Yeast Present in Kombucha

Yeast species may vary between Kombucha products, particularly those from different regions due to environmental factors such as geographic and climatic conditions, and even contamination between starter cultures [45,46]. Common yeasts isolated from Kombucha products produced in different regions and their characteristics (morphological and metabolic) are shown in Table 3 and Table 4 [24]. Generally, the yeast population outnumbers the bacteria present in Kombucha [47].

## 3. Isolation of AAB and Yeast from Kombucha

### 3.1. Isolation, Enumeration and Preservation of AAB

AAB can be isolated from fermented foods and beverages including vinegar, cider and lambic beers [60]. However, it is difficult to isolate AAB using commercial media as AAB are nutritionally demanding [61]. Although there are many growth media designed for the isolation of AAB which are based on their metabolism and nutritional requirements, other microorganisms can also grow on these media [62]. The carbon sources utilised are mainly D-mannitol and D-glucose, with different concentrations of ethanol and acetic acid added to the medium. Nitrogen sources are mainly peptone and yeast extract, with mineral salts including KH_2_PO_4_, Na_2_PO_4_ and MgSO_4_ added to the medium to aid the recovery of the AAB cells [61]. The composition of culture media commonly used for the isolation of AAB are shown in Appendix A and the references [28,60,63,64,65,66,67,68,69,70,71,72,73] are cited in the Appendix A.

Some AAB species are not culturable, therefore, alternative techniques such as real-time quantitative polymerase chain reaction (qPCR) are used to identify this population of AAB. Alternatively, epifluorescence staining techniques are also regarded as fast and simple methods for the enumeration of total viable/ non-viable AAB [74].

The preservation of AAB cultures is commonly achieved by sub-culturing, and storage under mineral oils, freeze-drying and cryopreservation, with frozen cultures stored at temperatures between −70 °C and −150 °C. Cell damage caused by ultra-low temperatures can be prevented by adding cryoprotectants such as glycerol (10–25%) and dimethyl sulfoxide (DMSO) (5%). However, glycerol is not suitable for the AAB that form cellulose structures such as *K. xylinus*. In these instances, DMSO is preferable as it can maintain stability and high viability without affecting the cellulose structure [75].

### 3.2. Isolation and Enumeration of Yeast

Conventional enumeration of yeast is carried out by spread plating as this technique allows microorganisms to be exposed to oxygen and avoids stress caused by hot or warm culture medium [76]. There are also numerous commercial media available for the isolation and cultivation of yeast from foods including Kombucha (Appendix A) [77,78]. These media provide the basic nutrients to support the growth of yeast and lower the pH to inhibit the growth of bacteria and moulds. Antibiotics such as penicillin can also be added to prevent the growth of bacteria and moulds [77].

## 4. Phenotypic Characterisation and Identification of Acetic Acid Bacteria and Yeast from Kombucha

### 4.1. Phenotypic Characterisation of AAB

The traditional classification of AAB species includes differentiation by cellular morphology, flagellation, and physiological and biochemical properties [79]. Colony examination involves size, form, elevation, and colour. The cellular morphology of bacteria includes the cell shape, size and response to Gram reaction, motility, spore-forming and cellular arrangement [80].

AAB are commonly Gram-negative, aerobic rods or ellipsoidal non-spore forming cells, which can be single, in pairs, or clusters. Cell sizes range from 0.4 to 1.0 μm in width and 0.8 to 4.5 μm in length [81]. These bacteria are mostly catalase-positive and oxidase negative. The optimum growth temperature varies from 25 °C to 30 °C, however, some thermotolerant strains can grow at temperatures up to 42 °C [21], and some AAB can grow under acidic conditions (e.g., pH 3.5) [60].

The classification of AAB comprises five chemical properties which are: catalase positive, oxidation of ethanol to acetic acid, over-oxidation of ethanol to water and CO_2_, oxidation of lactate to CO_2_ and water, ketogenesis from glycerol and hydrolysis of D-glucose to different acids [82]. Initially, AAB were classified into *Acetobacter* and *Gluconobacter.* The genus *Acetobacter* has peritrichous flagella and can oxidise acetate and lactate. In contrast, the genus *Gluconobacter* lacks the ability to oxidise acetate and lactate, however, they can oxidise D-glucose to 2-ketogluconate and 5-ketogluconate. The main difference between the genera *Acetobacter* and *Gluconobacter* is the presence of ubiquinone 9 in the genus *Acetobacter*, while ubiquinone-10 is present in the genus *Gluconobacter* [83]. In 1997, a new genus, *Gluconacetobacter*, was reported with partial 16 ribosomal sequencing techniques and the species containing coenzyme Q10 [68]. Later, another new genus, *Komagataeibacter*, was introduced based on 16S rRNA gene sequencing, phenotypic properties and different morphology to *Gluconacetobacter* [60]. The 11 species from the genus *Gluconacetobacter* were reclassified as *Komagataeibacter* based on their phenotypic and genotypic characteristics. Compared with *Gluconacetobacter*, the species of genus *Komagataeibacter* are not motile, and unable to produce a water-soluble brown pigment on glucose peptone yeast extract and calcium carbonate medium. The genus *Gluconacetobacter* can produce 2,5-diketo-D-gluconate but not *Komagataeibacter*. All AAB can oxidise sugars such as glucose, fructose, galactose, mannose, ribose and xylose through the cytoplasmic hexose monophosphate pathway [65]. Furthermore, AAB can oxidise sugar alcohols such as glycerol and convert them to dihydroxyacetone (DHA). Additionally, the ability to produce a cellulose structure is mainly found in the genera *Gluconacetobacter* and *Komagataebacter*; these differential properties help to distinguish AAB at the genus or species level. The main biochemical characteristics of the genera AAB applied in the food industry are shown in Table 5.

### 4.2. Phenotypic Characterisation and Identification of Yeast from Kombucha

The phenotypic identification of yeast from Kombucha is mainly achieved by morphological, and physiological tests and the use of rapid commercial yeast identification kits such as ID 32 C [85]. Traditionally, it is time-consuming to conduct physiological (identification) tests at the species level. The morphological characteristics of yeast cells are important for their identification as they may be produced by different reproduction modes as shown in Table 6 [86]. For example, the budding formation has been observed from yeast species, *Z. kombuchaensis* isolated from a dried tea fungus of Russian Kombucha [59]. Cellular morphological tests are usually obtained using the wet-mount method, by suspending the culture in saline and mixing it with dyes such as India ink, lactophenol cotton blue, calcofluor white or methylene blue staining [87].

Physiological and biochemical tests (Table 7) can be conducted using representative purified yeast to characterise the species present in food samples including Kombucha [77,79]. Kurtzman, Robnett and Basehoar-Powers conducted several nitrogen assimilation and carbohydrate fermentation tests in conjunction with other growth tests to determine the phenotypic characteristics of five new ascosporogeneous strains belonging to *Z. kombuchaensis*, which were isolated from Kombucha [59].

#### Identification of Yeast Using Commercial Kits

Several commercial identification systems based on conventional carbon fermentation or nitrogen assimilation reactions have been designed for rapid and accurate identification of food-related yeast in combination with morphological characterisation [77]. The ability of yeast to grow on different carbon and nitrogen sources can be determined by the formation of turbidity or colour changes in the presence of a pH indicator [85]. A description of common commercial systems such as API 20 C suitable for use in identifying yeast from Kombucha is summarised in Table 8.

Recently, several commercial kits have been successfully used to characterise yeast isolated from Kombucha. For example, the API 32 C system was used to characterise the carbohydrate assimilation pattern of the yeast strain, *Lanchacea fermentati* isolated from Kombucha [58]. Three different yeast species, *C. famata*, *C. krusei* and *C. magnoliae* from Kombucha SCOBY were identified using the API 20 C kit [57]. Commercial databases (Table 8) also contain yeast species isolated from Kombucha samples; hence, these commercial yeast identification systems may also be applied in the identification of yeast isolated from Kombucha by conducting different carbohydrate and nitrogen assimilation tests and comparing the results to the database.

## 5. Genotypic Identification of AAB and Yeast from Kombucha

### 5.1. Genotypic Identification of AAB from Kombucha

It is difficult to identify AAB to species levels using only phenotypic characteristics [61]. Compared with conventional biochemical and physiological identification, molecular techniques are generally more reliable and rapid [96]. Several DNA sequence-based techniques involving DNA extraction and polymerase chain reaction (PCR) have been widely applied to identify AAB to genera, species or strain levels by comparing with reference strains [97]. Commonly used DNA-based techniques applied for the identification of AAB are shown in Table 9.

Methods commonly used to discriminate AAB to species level include polymerase chain reaction-restriction fragment length polymorphism (PCR-RFLP) of the 16S rRNA genes or 16–23S internal transcribed spacer (ITS), PCR amplification and direct sequencing of 16S rDNAs and 16S–23S ITS, denaturing gradient gel electrophoresis (DGGE) of partial 16S rRNA gene, and real-time PCR (qPCR) [97]. The PCR-RFLP method involves amplification of the targeted regions of the 16S rRNA gene followed by digestion of the PCR products with restriction enzymes. The resultant DNA fragments are then separated by polyacrylamide or agarose gel electrophoresis. AAB isolates can be identified by comparing their restriction profiles with those of the reference strains [98]. Terminal restriction fragment length polymorphism (T-RFLP) is a modified version of PCR-RFLP, and it involves the use of fluorescent-labelled primers for detecting characteristic restriction fragments. T-RFLP has been used to determine the dynamic changes of AAB during the Kombucha fermentation process, and *Komagataeibacter* was revealed to be the dominant genus in both the tea fungus and the broth [17]. DGGE is commonly used to determine the diversity or species composition of AAB from fermented foods such as rice vinegar [62]. The principle of this technique is to separate DNA fragments of the same length on the basis of their differences in melting point, which is determined by the nucleotide sequence of the DNA molecule [97]. Thus, DGGE has a higher resolution than RFLP or T-RFLP in the discrimination of closely related AAB species.

Real-time PCR, also known as quantitative PCR (qPCR), is a well-established method that has been successfully used in the detection and quantification of AAB present in Kombucha. During each cycle of the qPCR reaction, fluorescence released by DNA-binding fluorescent dye or oligonucleotide probes is automatically measured as a proxy of the amplified DNA [99]. The amount of target gene (i.e., 16S rRNA) in the sample is then determined by calculating the cycle threshold (Ct), which is positively correlated with colony forming unit (CFU). qPCR is thus suitable for measuring AAB abundance in commercial Kombucha SCOBY [100].

To genetically validate the taxonomic identity of an AAB isolate, the full-length 16S rRNA gene is normally amplified by PCR and subsequently sequenced using the classic method of Sanger sequencing. However, along with the technical advancement of next-generation sequencing (NGS), the Illumina MiSeq has become a popular and powerful technique for determining microbial community structure and species composition. Briefly, total DNA is extracted from a Kombucha sample and then subjected to PCR amplification using general primers that target the V1–V3 (or V3–V4) region of the 16S rRNA gene [101,102,103]. The resultant amplicons are then sequenced using the Illumina MiSeq platform. An operational taxonomic unit (OUT) is assigned at a given level of sequence similarity (e.g., 97%). Moreover, the NGS technology has recently been extended beyond the analysis of targeted rRNA genes, and shotgun metagenomic sequencing enables the analysis of whole genomic DNAs that are extracted from a particular sample [104]. For example, Arika and other researchers analysed the microbial communities of homemade Kombucha fermentations using a combination of metagenomic sequencing and 16S rRNA amplicon sequencing [105]. Both methods consistently revealed that *Komagataeibacter* was the dominant bacterial genus. Interestingly, a search of secondary metabolite genes in the metagenome-assembled genomes identified novel gene clusters for bacteriocin production. Bacteriocins are a group of small antimicrobial peptides against closely related bacteria. The finding may partially explain the antimicrobial properties of Kombucha.

For identification to strain level, four PCR-based fingerprinting techniques are currently available: random amplification of polymorphic DNA (RAPD), amplified length fragments polymorphism (ALFP), enterobacterial repetitive intergenic consensus-PCR (ERIC-PCR), and repetitive extragenic palindromic PCR (REP-PCR) [97]. The RAPD technique involves the amplification of random regions of genomic DNA with a short (~10 nucleotides) single arbitrary primer under low annealing temperatures [106]. RAPD has been successfully used to identify the dominant bacterial composition of both tea fungus and the Kombucha broth [107]. In contrast, the ALFP technique selectively amplifies a subset of digested DNA fragments to generate a unique restriction profile for each AAB genome. ALFP involves the use of a pair of specifically designed primers, which consists of three parts: a core sequence, a restriction site sequence, and a 3′ selective sequence [97]. Both REP- and ERIC-PCR target repeat sequences that are commonly found in bacterial genomes. Both techniques can type AAB isolates to strain level, but ERI-PCR is more suitable for AAB due to its higher accuracy [108]. Interestingly, REP- and ERIC-PCR can be used in combination to increase the sensitivity of these two fingerprinting techniques, as shown in a case study with cellulose-forming AAB isolates from Kombucha [109]. Molecular techniques commonly applied in the identification of the Kombucha bacterial community are summarised in Table 10. The specific organisms and primers are linked to the work in reference in Table 10.

### 5.2. Genotypic Identification of Yeast

Different rapid commercial kits such as API 20 C and API 32 C are convenient for yeast identification, however, they are associated with minor differences in biochemical profiles due to variabilities in test conditions. Results from commercial kits need to correlate to morphological observations to identify the yeast at the species level. For instance, the species from *Dekkera* are the anamorphs of *Brettanomyces* and they are deficient in sexual characteristics. Thus, the biochemical profiles of these fungi with limited morphological features are not stable and therefore difficult to differentiate [111]. Consequently, accurate identification of yeast strains often needs a combination of conventional biochemical methods and modern molecular biology techniques.

Genotypic identification of yeast is mostly focused on sequence variations in the ribosomal DNA (rDNA) region, which includes 18S, 5.8S, and 26S rRNA genes separated by two internal transcribed spacers (ITSI and ITS2). Several universal primers have been designed for genus- or species-specific identification and the D1/D2 domain of 26S rDNA is the most popular target (Table 11). Yeast strains that differ by more than 1% in the 26S rDNA D1/D2 region are generally considered distinct species [112]. As mentioned above, high-throughput sequencing techniques such as Illumina MiSeq have also been used to examine the yeast communities in Kombucha, and the analysis can target either specific rRNA genes or the whole metagenome [102]. The relative abundance of yeast in Kombucha can be estimated using the real-time PCR method with one pair of specifically designed primers, which amplify a 124 bp region in the variable D1/D2 domain of the 26S rRNA gene [113]. Finally, it should be noted that other PCR-based techniques such as AFLP, RAPD and REP-PCR have been successfully developed for genetic identification of yeast at the sub-species or strain levels [112].

## 6. Conclusions and Future Research on Kombucha

Kombucha is a refreshing ‘live’ fermented beverage and its popularity is partially derived from this characteristic. While previous studies have shown that phenotypic identification provides useful metabolic characteristics of the microbes in Kombucha, molecular techniques, mostly based on PCR, can provide more accurate, rapid and reliable identification of the microbiological composition of AAB and yeast at different phylogenetic classification levels. The microbial composition of the Kombucha starter culture is diverse and is largely undefined. Further, the kinetic growth pattern of dominant fermenting microbes during fermentation are poorly described. Detailed information on the dominant bacteria and yeast responsible for the fermentation of Kombucha would be useful for better control of commercial fermentation processes during the production of safe, high-quality products. This review has highlighted the need for more information on the systematic identification methods of dominant yeast and AAB in commercial Kombucha beverages. In addition, although the presence of live cultures in Kombucha has been associated with its probiotic potential, this has been scantly reported. Therefore, more studies are needed to fill this gap in the literature.

## Figures and Tables

**Figure 1 foods-11-03456-f001:**
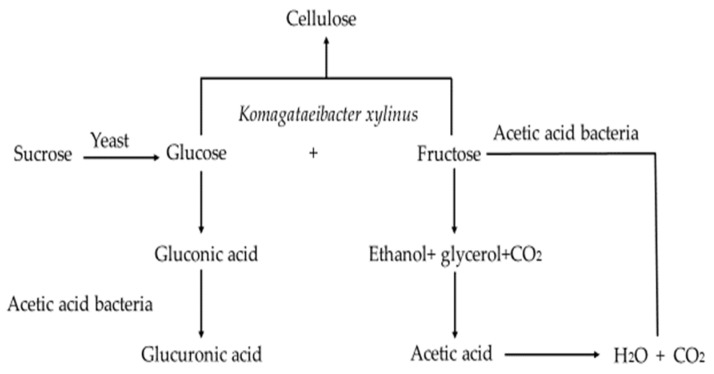
Metabolic activity of yeast and AAB during Kombucha fermentation. Adapted from Ref. [18].

**Table 1 foods-11-03456-t001:** Commercial Kombucha products sold in New Zealand.

Brand	Origin	Packaging	Flavour(s)	Storage Conditions
Remedy	Australia	330 mL (glass bottles) 250 mL (cans) 1.25 L (plastic bottles)	Ginger lemon, Raspberry Lemonade, Cola, Wildberry, Mango Passion, Passionfruit, Apple Crisp, Peach	Chilled/ambient temperature
Batchwell	New Zealand	375 mL (glass bottles)	Braeburn apple, Pineapple and Ginger, Motueka hops, Ginger and Turmeric, Early grey	Chilled
Daily Organics	New Zealand	200 mL (glass bottles) 1 L (glass bottles)	Original Kombucha, Chai spices and ginger, Lemon and ginger	Chilled
LO BROS	New Zealand	330 mL (glass bottles) 250 mL (cans) 750 mL (glass bottles)	Feijoa, Raspberry and lemon, Orange and mango, Ginger and lemon, Mango, Ginger and turmeric, Blueberry, Passionfruit, Cola, Ginger beer, Lemon lime and bitters, Pineapple and lime	Chilled/ ambient temperature
MAMA’S Brew Shop	New Zealand	375 mL (glass bottles) 330 mL (cans)	Lemongrass and ginger, Lavender and hibiscus	Chilled
Karma Drinks	New Zealand	330 mL (glass bottles)	Lemon and ginger, Raspberry and lemon, Mango and passionfruit, Cherryand berry	Chilled
Good Buzz	New Zealand	250 mL (cans) 328 mL (glass bottles) 375 mL (glass bottles) 888 mL (glass bottles)	Passionfruit and guava, Blueberry and peach, Pineapple and mango, Feijoa, Raspberry and lemon, Mango, Gisborne lemon and Manuka leaf, Hawkes Bay, peach and kawakawa	Chilled or ambient temperature

**Table 2 foods-11-03456-t002:** LAB isolated from Kombucha.

Species	Region	Reference
*Lacticaseibacillus casei*	NS	[35]
*Lactiplantibacillus plantarum*	China	[38]
*Lactobacillus nagelii*	United State of America (USA)	[39]
*Lactobacillus rhamnosus*	NS	[5]
*Lactobacillus mali*	USA	[39]
*Pediococcus (P.) pentosaceus*	Romania	[40]
*P.acidiliactici*	Romania	[36]

NS = not specified.

**Table 3 foods-11-03456-t003:** Common yeast species isolated from Kombucha and their metabolic characteristics.

Species	Morphology	Characteristics
*Zygosaccharomyces (Z.) bailii* [1]	White to cream colonies with brownish top, cylindrical or ellipsoidal shape, (3.5–6.0) × (4.5–11.5) μm in size	Tolerant to organic acids, Forms acetic acid, heat tolerance < 75 °C Growth pH > 2 and < 7 [48]
*Zycosaccharomyces (Z.) rouxii*	White to cream smooth colonies, round or oval shape	High osmotic stress and salt/sugar tolerant, grows under low oxygen and low water activity [49]
*Schizosaccharomyces (S.) pombe*	Cream to tan, butyrous colonies, rod-shaped	Can convert malic acid to ethanol, high resistance to low water activity, low pH and wide range of temperature environments, highly sugar content tolerant [50]
*Saccharomycodes (S.)* *ludwigi* [10]	Cream, butyrous colonies, elongated shape, and swelling in the centre	Resistant to pressurised carbon dioxide, high sugar tolerant [51]
*S. cerevisiae* [27]	White to cream, butyrous colonies, spherical or ovoid shape, 2.5–10.0 µm (diameter)	Can convert glucose to ethanol, high ethanol tolerance, rapid fermentation rate [52]
*Brettanomyces (B.)* *bruxellensis* [24,47]	Distinctive elongated shape, 2.5–10.0 µm (diameter)	Can produce high amounts of acetic acid and ethanol under aerobic conditions, high ethanol concentration (up to 15%), able to grow under low pH and oxygen environment, high efficiency to utilise nitrogen sources [53]

**Table 4 foods-11-03456-t004:** Yeast isolated from Kombucha in different regions.

Species	Country	References
*Brettanomyces (B.*) *anamalus*	New Zealand	[28]
*B. lambicus*	Germany	[54]
*B. custerisii*	Germany	[54]
*B. intermedius*	NS	[24]
*B. claussenii*	NS	[24]
*Candida* (*C.*) *albican* *C. colleculosa* *C. kefir* *C. krusei*	Japan Saudi Arabia Saudi Arabia Saudi Arabia	[24] [55] [55] [55]
*C. guilliermondii* *C. obtuse* *C. stellata*	Japan/Saudi Arabia Formosa Australia	[56] [46] [46]
*C. famata*	Indonesia	[57]
*C. magnoliae*	Indonesia	[57]
*Dekkera (D.) bruxelensis*	New Zealand	[28]
*Hanseniaspora (H.) valbyensis*	New Zealand	[28]
*Lachancea (L.) fermentati*	USA	[58]
*Kloeckera (K.) apiculata* *Kluyceromyces (K.) africanus*	NS NS	[24] [24]
*Pichia* (*P.*) *fermentans*	NS	[4]
*P. membranefaciens*	NS	[26]
*P. kudriavzevii*	New Zealand	[28]
*Torulaspora (T.) delbrueckii*	Australia	[46]
*Torulopsis (T.)famata*	Japan	[56]
*Zygosaccharomyces (Z.) kombucahensis*	Russia	[59]

Note: NS = not specified.

**Table 5 foods-11-03456-t005:** Differential characteristics of the genera *Acetobacter*, *Gluconacetobacter*, *Gluconobacter* and *Komagataeibacter* commonly associated with food.

Characteristics	*Acetobacter*	*Gluconacetobacter*	*Gluconobacter*	*Komagataeibacter*
Cell shape	Ellipsoidal to rods	Ellipsoidal to rods	Ellipsoidal to Rods	Coccoid to rods
Cell size (μm)	0.4–1.0 × 1.0–3.0	0.5–0.9 × 1.0–2.0	0.6–1.0 × 1.0–3.0	0.6–0.8 × 1.0–3.0
Colony appearance	Creamy to brown	Light brown to brownish	Smooth, entire, shiny, white, pink or brown	Raised, convex to umbonate, smooth to rough, entire to irregular
Catalase	+	+	+	+
Gram staining	Gram-negative	Gram-negative	Gram-negative	Gram-negative
Oxidase	−	−	−	−
Motility	Motile or non-motile	Motile or non-motile	Non-motile	No
Flagellation	Peritrichous	peritrichous	polar	No
Oxidation of ethanol to acetic acid	+	+	+	+
Oxidation of acetic acid to CO_2_ and water	+	+	−	+
Oxidation of lactate/acetate	+	+	−	+
Production of cellulose	−	±	−	±
Growth on 0.35% acetic acid	+	+	+	+
Growth in the presence of 1% KNO_3_	−	−	−	ND
Growth on methanol	±	−	−	ND
Growth in 30% D-glucose	−	±	±	ND
Ketogenesis (dihydroxyacetone) from glycerol	+	+	+	+
Acid production from: Glycerol D-Mannitol Raffinose Sorbitol	± − −	+ − − −	+ + − +	ND − ND −
Production of DHA from glycerol	±	±	+	+
Production of levan-like polysaccharide from sucrose	±	−	−	−
Ubiquinone type	Q9	Q10	Q10	Q10
G + C content (mol %)	50.5–60.3	55–67	52–64	56–64
Sources	Flowers, fruits, palm wine, vinegar, kefir	Rhizosphere of coffee plants, roots and stem of sugar cane	Strawberry, grape and spoiled jackfruit and sugar-rich environments	Kombucha, vinegar, wine vinegar

Source: [20,60,66,84]. Note: “+” 90% or more strains shown positive results; “−” 90% or more strains shown negative results; ND: No data; “±”: strains are positive or negative.

**Table 6 foods-11-03456-t006:** Morphology of yeast isolated from Kombucha.

Different Reproduction Mode of Yeast	Morphology Characteristics of Yeast
Vegetative or asexual reproduction	Budding: new cell is produced on the surface of parent cell and then separate Fission: an asexual cell is produced by a septum grown inward from cell wall to halve the long axis of the cell. Blastoconidiation: a mother cell of stalk-like tubular sterigmata produce a terminal conidium
Sexual reproduction in ascomycetous yeast	Parent cell-bud conjugation Gametangial conjugation Heterothallism conjugation Conjugation between hyphae
Sexual reproduction in basidiomycetous yeast	Budding haplophase Dikaryotic hyphal phase or self-spore forming diplophase

**Table 7 foods-11-03456-t007:** Physiological and biochemical tests used for the characterisation of yeast.

Physiological Test	Biochemical Test
Assimilation of carbon and nitrogen sources	Diazonium Blue B reaction
Fermentation of carbohydrates	Urease test
Growth at different temperature	
Growth in vitamin-free medium	
Growth in high osmotic pressure condition	
Starch hydrolysis activity	

**Table 8 foods-11-03456-t008:** Summary of tests and accuracy of common commercial yeast identification systems.

Commercial System	Description of Tests and Controls Included in the Kits	Incubation Condition	Accuracy (%)	Reference
API 20 C	19 carbon assimilation test and 1 control test in 20 strips	30 °C for 72 h	98.9	[88]
API Candida	5 carbohydrate and 7 enzymes colorimetric test in 10 strips	35 °C for 18–24 h	97.4	[89]
API 32 C	29 assimilation tests (carbohydrate, organic acids, and amino acids); 1 negative control, 1 susceptibility test (cycloheximide) and 1 colorimetric test (esculin) in 32 wells. Includes 63 different species in database	30 °C for 48 h	92	[90]
Auxacolor system	13 carbohydrate tests with bromocresol purple, test for cycloheximide resistance and phenoloxidase production in 16 wells.	37 °C for 48 h	79.4	[91]
RapID Yeast Plus system	5 carbon assimilation tests and 13 enzymatic hydrolysis substrate tests	30 °C for 4 h	96	[92]
The Uni-Yeast-Tek (UYT) system	7 carbon assimilation tests, urease, Nitrate and corn meal with Tween 80 agar	22–26 °C for 2–10 days	99.8	[93]
MicroScan yeast identification panel	13 aminopeptidase, 3 carbohydrates, 9 glycosidase, phosphatase and urease tests.	37 °C for 4 h	86.9	[94]
VITEK 2 YST	4 aminopeptidase, 25 carbohydrate, esculin, 3 glycosidase, nitrate, 2 nitrogen, 9 organic acid, and urea tests	35 °C for 18 h	94.8	[95]

**Table 9 foods-11-03456-t009:** Commonly used molecular techniques applied in AAB identification and genotyping.

Technique	Level	Advantage	Disadvantage
PCR-RFLP	Species	Rapid, cheap, and easy to set up; suitable for genotyping of AAB isolates	Difficult to identify small insertions and expensive, unable to discriminate closely related species
DGGE	Species	Rapid and cost-effective; suitable for estimation of AAB diversity	Cannot discriminate closely related species.
Real-time PCR	Species	Rapid, reliable and quantitative; suitable for comparing microbial abundance.	Complex and expensive
RAPD	Strain	Quick and simple	Low reproducibility, as the quality and concentration of template DNA influence the results
ALFP	Strain	Can be used for any DNA samples of any origin, and reveal multiple polymorphic bands in one lane.	Complex and sensitive

**Table 10 foods-11-03456-t010:** Molecular techniques commonly used for AAB identification.

Method	Microorganism	Primer Sequence (5′–3′)	Reference
16S rRNA, Sanger sequencing	*Gluconacetobacter*;	27F, AGAGTTTGATCMTGGCTCAG	[107]
	*Komagataeibacter*;	1494R, TGACTGACTGAGGYTACCTTGTTACGACTT	
	*Gluconacetobacter*;	fD1, CCGAATTCGTCGACAACAGAGTTTGATCCTGGCTCAG	[29]
	*kombuchae* sp. nov.;	rD1, CCCGGGATCCAAGCTTAAGGAGGTGATCCAGCC	
	*Acetobacter aceti*;	16S d, GCTGGCGGCATGCTTAACACA	[96]
		16S r, GCAGGTGATCCAGCCGCA	
16S rRNA V1-V3 region, Illumina MiSeq	Bacterial communities	Forward, TCGTCGGCAGCGTCAGATGTGTATAAGAGACAG	[108]
		Reverse, GTVTVGTGGGCTCGGAGATGTGTATAAGAGACAG	
16S rRNA V3-V4 region, Illumina MiSeq	Bacterial communities	Forward, CCTACGGGNGGCWGCAG	[102] [109]
		Reverse, GACTACHVGGGTATCTAATCC	
16S rRNA, Real-time PCR	Bacterial abundance	926f, AAACTCAAKGAATTGACGG	[100]
		1062r, CTCACRRCACGAGCTGAC	
16S rRNA, T-RFLP	Bacterial communities	27F, AGAGTTTGATCMTGGCTCAG	[17]
		1525R, AAGGAGGTGATCCAGCC	
RAPD	*Komagataeibacter* spp.	M13, GAGGGTGGCGGTTCT	[104]
AFLP	*Komagataeibacter rhaeticus*	A03, GACTGCGTACAGGCCCCG	[110]
		T03, CGATGAGTCCTGACCGAG	
REP-PCR	*G. oxydans*; *A. aceti*	REPIR-I, IIICGICGICATCIGGC	[105]
		REP2-I, ICGICTTATCIGGCCTAC	
ERIC-PCR	*G. oxydans*; *A. aceti*	ERIC1R, ATGTAAGCTCCTGGGGATTCAC	[105]
		ERIC2, AAGTAAGTGACTGGGGTGAGCG	
Shotgun metagenomic sequencing	Bacterial and fungal communities		[101] [102]

**Table 11 foods-11-03456-t011:** Molecular techniques used for the identification of yeast present in Kombucha.

Method	Microorganism	Primer Sequence (5′–3′)	Reference
26S rDNA, D1/D2 domain	*Brettanomyces/Dekkera*; *Pichia*; *B. bruxellensis*;	NL1, GCATATCAATAAGCGGAGGAAAAG	[107] [104]
	*D. bruxellensis*; *Hanseniaspora (H.) uvarum*	NL4, GGTCCGTGTTTCAAGACGG	[109]
26S rDNA, D1/D2 domain	*D. bruxellensis*; *D. anaomala*; *Z. bailii*; *H. valbyensis*	NL1, GCATATCAATAAGCGGAGGAAAAG	[108]
		NL4, GGTCCGTGTTTCAAGACGG	
18S rDNA, D1/D2 domain	*Z. kombuchaensis* sp.	NS-1, GTAGTCATATGCTTGTCTC	[59]
		NS-8A, CCTTCCGCAGGTTCACCTACGGAAACC	
ITS, Illumina MiSeq	*Z. bailii*	ITS1F, CTTGGTCATTTAGAGGAAGTAA	[102]
		ITS2R, GCTGCGTTCTTCATCGATGC	
Real-time PCR	*Brettanomyces*	Yeast-F, GAGTCGAGTTGTTTGGGAATGC	[100]
		Yeast-R, TCTCTTTCCAAAGTTCTTTTCATCTTT	
PCR-ITS RFLP	*D. bruxellensis*;	ITS1, TCCGTAGGTGAACCTGCGG	[47]
		ITS4, TCCTCCGCTTATTGATATGC	

## Data Availability

Not applicable.

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
