# Peer review of "Kombucha: Production and Microbiological Research"

_foods, 2022, doi:10.3390/foods11213456_

Round 1

Reviewer 1 Report

The article is devoted to the elucidation of the microbial content of kombucha, a legendary fermented drink with amazing properties. The manuscript contains a lot of detailed information, but before that, several mandatory improvements are needed.

1.     This review certainly must include a paragraph describing the beneficial properties of kombucha for human health. There are works claiming that this drink has anticarcinogenic, antihypertensive, antidiabetic, and hepatoprotective effects (https://www.sciencedirect.com/science/article/pii/B9780128152607000080). The authors should describe these effects and whether are they related to the presence of probiotic bacteria in this drink. Otherwise, the authors have to show their opinion if the drink has contradictory properties, and in which cases its consumption is even not recommended (https://www.sciencedirect.com/science/article/pii/S1047279718307385).

2. Since the authors discuss genetic methods for identifying the different microbial species, the metagenomic analyses of kombucha samples should be described as well. There is a lot of information about the Next-generation sequencing of kombucha samples recently:

https://ift.onlinelibrary.wiley.com/doi/10.1111/1750-3841.14992

https://link.springer.com/article/10.1007/s13197-022-05476-3

3. Attention must be paid to accurate microbial taxonomy. Lactobacillus plantarum should be changed to Lactiplantibacillus plantarum, Lactobacillus casei should be Lacticaseibacillus casei, Komagataeibacter xylinum is Komagataeibacter xylinus. All microbial names should be in italics.

4. The information in Tables 5 and 6 should be given in Supplements as it is informative but not of interest to the general public. Table 7 is absolutely redundant and Table 8 needs shortening and reformatting.

5. Access links should be added to the references.

Author Response

Response to Reviewer 1

Title:  Kombucha: Production and Microbiological Research

The article is devoted to the elucidation of the microbial content of kombucha, a legendary fermented drink with amazing properties. The manuscript contains a lot of details information, but before that, several mandatory improvements are needed.

  1. This review certainly must include a paragraph describing the beneficial properties of Kombucha for human health. There are works claim that this drink has anticarcinogenic, antihypertensive, antidiabetic, and hepatoprotective effects. (https://www.sciencedirect.com/science/article/pii/B9780128152607000080).

The authors should describe these effects and whether are they related to the presence of probiotic bacteria in this drink. Otherwise, the authors have to show their opinion if the drink has contradictory properties, and in which cases its consumption is even not recommended. (https://www.sciencedirect.com/science/article/pii/S1047279718307385),

Response/Comment: A paragraph has been added on the health benefits and potential risks of Kombucha in the introduction section and text written in blue (Ln 82-Ln 99).

  1. Since the authors discuss genetic methods for identifying the different microbial species, the metagenomic analyses of kombucha samples should be described as well. There is a lot of information about the Next-generation sequencing of kombucha samples recently:

https://ift.onlinelibrary.wiley.com/doi/10.1111/1750-3841.14992 https://link.springer.com/article/10.1007/s13197-022-05476-3

Response/Comment: The discussion about the metagenomic has been added in Ln354-Ln 358 and the text coloured blue.

  1. Attention must be paid to accurate microbial taxonomy. Lactobacillus plantarum should be changed to Lactiplantibacillus plantarum, Lactobacillus casei should be Lacticaseibacillus casei, Komagataeibacter xylinum is Komagataeibacter xylinus. All microbial names should be in italics.

Response/Comment: The microbial names have been updated and written in italics.

  1. The information in Table 5 and 6 should be given in supplements as it is informative but not of interest to the general public. Table 7 is absolutely redundant and Table 8 needs shortening and reformatting.

Response/Comment: Table 5 and Table 6 have been reformatted and moved to the Supplement section. Table 7 has been deleted. Table 8 has been shortened and reformatted.

  1. Access links should be added to the references.

Response/Comment: Where applicable, the links have been added at the end of the references.

Reviewer 2 Report

foods-1955037-peer-review-v1

This is interesting work, summarizing of information on Kombucha, and the different methods and approaches in the isolation and identification AAB, LAB and yeasts from Kombucha. In my opinion paper can be suggested for publication, but some adjustments, corrections, upgrade needs to be taken under consideration by the authors.

Please, all comments are mentioned on the pdf file as highlighted and sticky notes comments.

Author Response

Response to Reviewer 2

Title:  Kombucha: Production and Microbiological Research

This is interesting work, summarizing of information on Kombucha, and the different methods and approaches in the isolation and identification AAB, LAB and yeasts from Kombucha. In my opinion paper can be suggested for publication, but some adjustments, corrections, upgrade needs to be taken under consideration by the authors.

Response/Comments: The manuscript has been modified by track change according to the reviewer’s feedback. 

Reviewer 3 Report

This is a very good review paper. The Authors tried and successfully presented the central aspect of the preparation of kombucha beverages, with particular highlights on the microbiological profile of this fermented product.

Although the paper looks structured and good-organized, I think that some younger references need to be put on paper (published in the last 5 years), especially in part about the identification of microorganisms in kombucha.

Also, Tables 5 and 6 are not readable, so I suggest rewriting these tables. I am going to put some recommendations on how to do this. If Authors have a better idea to present so many compounds and different types, they can use it. everything is better than the present situation.

Medium

Components

(g/L) or (mL/L)

Refs

Glucose

Yeast extract

Calcium carbonate

Agar

Peptone

Acetic acid

Ethanol

Na2Po4 H2O

Citric acid x H2O

Mannitol

Malt extract

Vitamin free casamino acids

Bromocresol green

Sorbitol

Na2PO4

Clucose yeast extract carbonate (GYC)

100

10

20

15

/

/

/

/

/

/

/

/

/

/

/

[62]

Glucose yeast extract (GY)

20

10

/

20

/

/

/

/

/

/

/

/

/

/

/

[57]

 ...

 ...

 ....

 ...

Author Response

Response to Reviewer 3

Title:  Kombucha: Production and Microbiological Research

This is a very good review paper. The Authors tried and successfully presented the central aspect of the preparation of kombucha beverages, with particular highlights on the microbiological profile of this fermented product.

  1. Although the paper looks structured and good-organized, I think that some younger references need to be put on paper (published in the last 5 years), especially in part about the identification of microorganisms in kombucha.

Response/Comment: We agree. Four recent references have been added.

  1. Also, Tables 5 and 6 are not readable, so I suggest rewriting these tables. I am going to put some recommendations on how to do this. If Authors have a better idea to present so many compounds and different types, they can use it. everything is better than the present situation.

Response/Comment: We agree with Reviewer comment. The names of media have been shortened and the units in the table have been deleted. However, due to the high number of ingredients in the different media, we have retained the previous format and made some modifications according to the recommendations given by the reviewer.
